# Advances in *Azorella glabra* Wedd. Extract Research: In Vitro Antioxidant Activity, Antiproliferative Effects on Acute Myeloid Leukemia Cells and Bioactive Compound Characterization

**DOI:** 10.3390/molecules25214890

**Published:** 2020-10-22

**Authors:** Daniela Lamorte, Immacolata Faraone, Ilaria Laurenzana, Stefania Trino, Daniela Russo, Dilip K. Rai, Maria Francesca Armentano, Pellegrino Musto, Alessandro Sgambato, Luciana De Luca, Luigi Milella, Antonella Caivano

**Affiliations:** 1Laboratory of Preclinical and Translational Research, Centro di Riferimento Oncologico della Basilicata (IRCCS CROB), 85028 Rionero in Vulture, Potenza, Italy; ilaria.laurenzana@crob.it (I.L.); stefania.trino@crob.it (S.T.); 2Department of Science, University of Basilicata, V.le dell’Ateneo Lucano 10, 85028 Rionero in Vulture, Potenza, Italy; immafaraone88@gmail.com (I.F.); daniela.russo@unibas.it (D.R.); mariafrancesca.armentano@unibas.it (M.F.A.); 3Spinoff BioActiPlant s.r.l., University of Basilicata, V.le dell’Ateneo Lucano 10, 85028 Rionero in Vulture, Potenza, Italy; 4Department of Food BioSciences, Teagasc Food Research Centre Ashtown, D15KN3K Dublin, Ireland; Dilip.Rai@teagasc.ie; 5Hematology and Stem Cell Transplantation Unit, Centro di Riferimento Oncologico della Basilicata (IRCCS CROB), 85028 Rionero in Vulture, Potenza, Italy; p.musto@tin.it; 6Scientific Direction, Centro di Riferimento Oncologico della Basilicata (IRCCS CROB), 85028 Rionero in Vulture, Potenza, Italy; alessandro.sgambato@crob.it; 7Unit of Clinical Pathology, Centro di Riferimento Oncologico della Basilicata (IRCCS CROB), 85028 Rionero in Vulture, Potenza, Italy; luciana.deluca@crob.it (L.D.L.); antonella.caivano@crob.it (A.C.)

**Keywords:** *Azorella glabra* Wedd., *Azorella diapensioides* A. Gray, *yareta*, phytochemicals, terpenoids, antioxidant, acute myeloid leukemia, KG1, MV4-11, cytotoxic effect, apoptosis, cell cycle arrest

## Abstract

*Azorella glabra* Wedd. (AG) is traditionally used to treat gonorrhea or kidney’s problems. The antioxidant, antidiabetic, anticholinesterase and in vitro antitumor activities of AG extracts were recently reported. The aim of this work was to investigate anti-leukemic properties of AG chloroform fraction (AG CHCl_3_) and of its ten sub-fractions (I-X) and to identify their possible bioactive compounds. We determined their in vitro antioxidant activity using 2,2’-azino-bis(3-ethylbenzothiazoline-6-sulfonic acid) (ABTS), nitric oxide (NO) and superoxide anion (SO) assays, and their phytochemical profile by spectrophotometric and LC-MS/MS techniques. I-X action on two acute myeloid leukemia (AML) cell lines viability, apoptosis and cell cycle were evaluated by MTS, western blotting and cytofluorimetric assays. Different polyphenol, flavonoid and terpenoid amount, and antioxidant activity were found among all samples. Most of I-X induced a dose/time dependent reduction of cell viability higher than parent extract. IV and VI sub-fractions showed highest cytotoxic activity and, of note, a negligible reduction of healthy cell viability. They activated intrinsic apoptotic pathway, induced a G_0_/G_1_ block in leukemic cells and, interestingly, led to apoptosis in patient AML cells. These activities could be due to mulinic acid or azorellane terpenoids and their derivatives, tentatively identified in both IV and VI. In conclusion, our data suggest AG plant as a source of potential anti-AML agents.

## 1. Introduction

Plant extracts play an important role in antitumor drug discovery thanks to their large structural diversity and low toxicity towards normal cells. Of note, they not only can target different pathways, like proliferation and differentiation in malignant cells [1,2], but can also reduce the resistance against chemotherapies [3,4] and can exhibit a synergistic effect with conventional drugs [5,6].

The necessary steps to isolate the biologically active compounds from plants are extraction, pharmacological screening, isolation and characterization of bioactive compounds, toxicological and clinical evaluations [7]. Most studies reported the extraction of bioactive compounds from aerial part (i.e., stem, leaves and flowers) of plants [2,8,9,10]. The selection and the polarity of solvents used for the extraction are crucial, as the nature of the extracted compounds depends on them. Due the high complexity of phytochemicals, the use of solvents having increasing polarity is useful for highly valued separations [11].

Polyphenols, flavonoids and terpenoids are the main bioactive compounds of plant extracts. They show antioxidative, anti-inflammatory and antitumor properties, which makes them very interesting in several chronic disease associated with oxidative stress, including cancer. Terpenoids are the most extensively studied as anticancer agents thank to their ability to induce apoptosis, cell cycle arrest and tumor cell differentiation, and to inhibit angiogenesis, invasion and metastasis [12,13].

Plants of *Azorella* genus are commonly used as infuse in folk medicine in the Andean region of South America to treat a variety of ailments, such as colds, asthma, bronchitis and different conditions in which the main symptoms include inflammation and pain [14,15,16]. Although the antiproliferative and proapoptotic effects of *Azorella* genus have been proven in various tumors, including hematologic malignancies [14,17], little information is available for *Azorella glabra* (AG) Wedd., an endemic Bolivian species also known as *Azorella diapensioides* A. Gray and popularly called “*yareta*” [18]. AG is traditionally used to treat gonorrhea or problems of the kidney [19,20]. Recently, the antioxidant, antidiabetic and anticholinesterase activities of AG aerial parts extracts were described [10]. In addition, we previously reported that the chloroform fraction (CHCl_3_), obtained from AG ethanolic extract by liquid/liquid extraction, induces, in vitro, apoptosis and cell cycle arrest and inhibits cell migration of multiple myeloma (MM) cells [2]. Today, there are no additional studies reporting the cytotoxic properties of AG extracts on acute myeloid leukemia (AML).

AML is an aggressive and heterogeneous hematological malignancy characterized by an abnormal proliferation and differentiation of myeloid precursors and by their accumulation in the bone marrow (BM) and peripheral blood (PB) [21,22]. Different combinations of cytogenetic, epigenetic and molecular abnormalities are responsible of leukemic features [23,24]. AML shows a median age at diagnosis of about 70 years [25], but it also accounts for 15–20% of pediatric acute leukemia’s [26]. This neoplasm is associated with a five-year survival rate of 40–45% in young patients and less than 10% in the elderly patients [26]. Although novel strategies, including the refinements of conventional chemotherapies, hypomethylating agents and molecular targeted drugs, have been developed in recent years, resistance and relapse remain the main clinical problems in AML [1]. Thus, the development of novel drugs is need.

During the last years, research has focused on different plant extracts as possible source of antitumor agents for various types of hematologic malignancies, including AML [27,28,29].

In the present work, it was firstly reported the effect of AG CHCl_3_ fraction on AML viability, and then, the fractionation of AG CHCl_3_ to obtain less complex sub-fractions (I-X) enriched in bioactive compounds. Afterwards, since it is known that oxidative stress is the basis of tumors and that the antioxidant activity is related to phytochemical profile, we investigated the phytochemical profile, the in vitro antioxidant activity and the action of I-X sub-fractions on AML and healthy cell viability. Finally, we tentatively characterized the composition of the most bioactive sub-fractions by mass spectrometry.

## 2. Results

### 2.1. Viability Analysis of AML Cells Treated with AG CHCl_3_ Fraction

To verify whether AG CHCl_3_ fraction could alter the viability of AML cells, we treated two AML cell lines, KG1 and MV4-11, with different concentrations (10–25–50 μg/mL) of AG CHCl_3_ for 24 h and 48 h. AG CHCl_3_ induced a statistically significant reduction of both AML cell line viability (Figure 1). The reduction of AML cell viability was already evident at 24 h of treatment with 25 μg/mL for KG1 and with 50 μg/mL for MV4-11 cells.

Then, we calculated the EC_50_ of AG CHCl_3_ fraction observing that it was similar in both cell lines and it decreased from 24 h to 48 h (Table 1).

### 2.2. Fractionation, Phytochemical Profiles, and In Vitro Antioxidant Activity of AG CHCl_3_ Sub-Fractions

The AG CHCl_3_ fraction was fractionated in order to get sub-fractions enriched in bioactive compounds. We obtained ten sub-fractions (I-X). The phytochemical profile and the in vitro antioxidant activity of I-X sub-fractions were determined. All samples showed differences in the amount of polyphenols and terpenoids, except for I sub-fraction that contained only polyphenols (Figure 2a). In particular, X sub-fraction showed the highest Total Polyphenol Content (TPC) value (49.87 ± 0.36 mg GAE/g) (Figure 2a). Flavonoid compounds were present only in II and IX sub-fractions (85.59 ± 7.08, and 88.09 ± 3.54 mg QE/g, respectively), with lower values than parent AG CHCl_3_ fraction (121.11 ± 7.57 mg QE/g). VI, VII and VIII sub-fractions exhibited higher terpenoid content (840.56 ± 23.76, 1340.36 ± 9.90, and 1693.16 ± 49.50 mg LE/g, respectively) than AG CHCl_3_ fraction (733.51 ± 9.42 mg LE/g) (Figure 2a).

The antioxidant activity was evaluated by three complementary tests: 2,2-azino-bis(3-ethylbenzothiazoline-6-sulfonic acid) (ABTS), superoxide anion (SO), and nitric oxide (NO) assays. In ABTS assay, all samples presented a lower activity than the initial AG CHCl_3_ fraction (32.08 ± 0.02 mg TE/g; Table 2) indicating a possible synergistic effect of the whole phytocomplex. The sample ability to scavenge the biological SO and NO was expressed as concentration of the sample required to inhibit the activity of the radical by 25% (IC_25_) and results were compared with ascorbic acid. AG samples caused a dose-dependent inhibition in SO (Table 2) but not in NO (data not shown). In particular, I-II-V-VI-VIII-IX and X sub-fractions showed higher activity than ascorbic acid (IC_25_ of 261.48 ± 17.60 µg/mL); II and IX sub-fractions resulted the most active (Table 2). However, as the antioxidant measurement scale of each method is different, it is difficult to define the antioxidant capacity of samples. To overcome this problem, we calculated the statistical Relative Antioxidant Capacity Index (RACI), an arbitrary index that integrates the results obtained from ABTS and SO in vitro antioxidant assays together with TPC. RACI values evidenced as X, II, IX and V sub-fractions had the highest values, followed by the initial AG CHCl_3_ fraction, while VI-VII-I-IV-VIII-III sub-fractions presented the lowest RACI index and, therefore, a relative lack of antioxidant activity (Figure 2b).

### 2.3. Viability Analysis of AML and Healthy Cells Treated with AG CHCl_3_ Sub-Fractions

We treated KG1 and MV4-11 cells with I-X sub-fractions or with dimethyl sulfoxide (DMSO) vehicle control at different concentrations (10–25–50 μg/mL) for 24 h and 48 h. MTS assay showed that most of AG sub-fractions exhibited a dose and time dependent viability reduction of AML cells, higher than AG CHCl_3_ (Figure 3a,b). Of note, the effects of IV and VI sub-fractions were the strongest already after 24 h on both AML cell lines.

Consequently, we calculated the EC_50_ value for IV and VI sub-fractions (Table 3 and Appendix A). Both sub-fractions showed a similar EC_50_ value, ranging from about 9 to 35 μg/mL.

Likewise, to support their specific antitumor effect, we treated healthy peripheral blood mononuclear cells (hPBMCs) with different concentrations (10–25–50 μg/mL) of IV and VI sub-fractions for 24 h and 48 h. Interestingly, their action on hPBMC viability was negligible after treatment with 10 and 25 μg/mL (Figure 3c,d).

### 2.4. Evaluation of Apoptosis and Cell Cycle in KG1 and MV4-11 Cells Treated with AG IV and VI Sub-Fractions

To further investigate the viability reduction induced by IV and VI sub-fractions, we evaluated both apoptosis and cell cycle assays after 24 h and 48 h of treatment using, for each cell line, the IV and VI sub-fractions EC_50_ value calculated at 48 h (30 μg/mL of IV and VI sub-fractions for KG1, and 10 μg/mL of IV and VI sub-fractions for MV4-11 cells).

Apoptosis assay showed a significant increase of KG1 and MV4-11 apoptotic cells in a time dependent manner. The proapoptotic effect was already present after 24 h in both cell lines (Figure 4).

The occurrence of apoptosis was also confirmed by western blot analysis, investigating the expression levels of cPARP-1 and Bcl2 (Figure 5). IV and VI sub-fractions induced a significant increase of cPARP-1 and a decreased expression of Bcl2. The increase of cPARP-1 was more evident at 24 h than at 48 h of treatment in both cell lines, while the reduction of Bcl2 expression at 24 h slightly increased at 48 h of treatment in both AML cells.

Moreover, we observed a common trend of G_0_/G_1_ phase increase accompanied by a decrease of G_2_/M in both cell lines after treatment with both sub-fractions. In particular, regarding IV sub-fraction, the data was significant for MV4 cells (*p* ≤ 0.05 at 48 h), while it was not significant for KG1 where, in any case, a trend in this direction can be observed. For VI sub-fraction treatment, the G_0_/G_1_ phase increase was significant in both cell lines (*p* ≤ 0.05 on KG1; *p* ≤ 0.001 on MV4-11 at 48 h) and the decrease of G_2_/M was significant in MV4-11 cells (*p* ≤ 0.05 at 48 h) (Figure 6).

### 2.5. Evaluation of Apoptosis and Cell Cycle in Primary AML Cells Treated with AG IV and VI Sub-Fractions

We evaluated apoptosis and cell cycle in bone marrow mononuclear cells (BMMCs) collected from three AML patients treated with IV and VI sub-fractions (see materials and methods for patient characteristics). In particular, BMMCs were treated with 30 μg/mL of each fraction for 24 h and 48 h. IV sub-fraction induced an increase of apoptosis already at 24 h, which was maintained at 48 h in AML patient 1 and 3, while in AML patient 2 the weak induced apoptosis at 24 h was increased at 48 h (Figure 7a). In a similar way, VI sub-fraction induced an increase of apoptosis at 24 h and 48 h in primary AML cells from patient 1 and 3, while in AML cells from patient 2 the increase of apoptosis was observed only at 48 h (Figure 7b).

Cell cycle assay on primary AML cells did not show any difference between treated and control cells (data not shown).

### 2.6. Phytochemical Identification of AG CHCl_3_ IV and VI Sub-Fractions

To identify compounds in IV and VI sub-fractions, we performed the LC-MS/MS analysis in negative mode. We identified 5 more abundant picks (**1**–**5**) in IV sub-fraction and 7 (**1**–**7**) in VI sub-fraction (Figure 8). Due to the absence of literature studies to our knowledge and to the lack of available commercial standards, all compounds were tentatively identified (Table 4 and Table 5) through accurate masses and possible fragmentation patterns.

The compounds with *m/z* 333.20 ((**1**) and (**2**) in IV sub-fraction and 6 in VI sub-fraction) were tentatively identified as mulinic acid or 14*α*-hydroxymulin-12-en-11-one-20-oic acid or isomers (C_20_H_30_O_4_). Upon fragmentation by collision–induced dissociation (CID), these compounds produced the ions at *m/z* 289, 287, 271, 253, 229, 135, 113, 95, 87, 83, 69, 57. In particular, Areche et al., in 2019, reported the ion fragment at *m/z* 135 for 14*α*-hydroxymulin-12-en-11-one-20-oic acid, a mulinane diterpenoid [30]. The signal at 16.61 min in IV sub-fraction (**5**) with *m/z* 497.27 indicated the presence of an extra hexose sugar unit with respect to *m/z* 333.20 and it was putatively identified as mulinic acid hexoside derivative.

The compounds with *m/z* 331.19 ((**3**) in IV sub-fraction and (**7**) in VI sub-fraction) were tentatively identified as mulin-12-ene-11,14-dion-20-oic acid or isomers (C_20_H_28_O_4_). Upon fragmentation by CID, these compounds produced the ions at *m/z* 287, 269, 259, 243, 229, 215, 121, 111, 109, 96, 83, similar to reported by Wächter et al. in 1999 [31].

The compound with *m/z* 303.19 ((**4**) in IV sub-fraction) was tentatively identified as azorellanone [32] or dihydroazorellolide [33].

The signal at 10.76, 10.97 and 11.53 min in VI (**1**, **2** and **4**) with *m/z* 335.22 were putatively identified as 13,14-dihydroxymulin-11-en-20-oic acid or isomers. Upon fragmentation by CID, these compounds produced the ions at *m/z* 289, 273, 245, 235, 59 [34].

The compounds with *m/z* 349.23 (**3** in VI sub-fraction) and *m/z* 393.24 (**5** in VI sub-fraction) were tentatively identified as mulin-en-dion-oic acid derivative and dihydroxymulinic acid derivative, respectively.

Of note, compounds with *m/z* 333 and 331 were present in both sub-fractions.

## 3. Discussion

Phytoextracts and their derived compounds are being increasingly recognized as complementary and adjuvant agents for cancer treatments [35].

Most studies reported the extraction of bioactive compounds from aerial part (i.e., stem, leaves and flowers) of plants, including for that belonging to the Apiaceae family as *Azorella glabra* (AG) [2,8,9,10]. In addition, in this work, AG samples were extracted from the aerial parts of plant by dynamic maceration with 96% ethanol; then, the ethanol extract was separated by liquid/liquid extraction using solvent with increasing polarity obtaining six fractions, including the chloroform one (AG CHCl_3_) [2]. The selection and the polarity of solvents used for the extraction are crucial, as the nature of the extracted compounds depends on them. Finally, AG CHCl_3_ was further fractionated by column chromatography using solvent with increasing polarity, and the obtained fractions were re-united on the basis of their thin-layer chromatography (TLC) profile. Column chromatography and TLC are commonly used for their simplicity, economy and availability in various stationary phases, such as silica, alumina, cellulose and polyamide. Furthermore, due the high complexity of phytochemicals, the use of solvents having increasing polarity phases is useful for highly valued separations [11].

Although the antitumor property of *Azorella* genus extracts were recently reported [14,34,36], AG Wedd. extracts have not yet been extensively explored. Additionally, there are few information about AG bioactive phytocompounds. To our knowledge, our recent work is the first study reporting the cytotoxic activity of AG extracts [2]. In particular, we showed that the AG CHCl_3_ fraction reduced cell viability, induced apoptosis and cell cycle arrest, and inhibited migration in MM cells.

In the present study, we proved the antitumor activity of the AG CHCl_3_ fraction in AML. Interestingly, we reported the ability of AG CHCl_3_ fraction to reduce cell viability of an erythroleukemia cell line (KG1) and of a monocytic leukemia cell line (MV4-11) (Figure 1). Subsequently, we further fractionated the above fraction and we analyzed the phytochemical profile, the in vitro antioxidant activity and the cytotoxic effects of its ten sub-fractions (I-X).

The phytochemical profile of AG samples showed differences in amount of polyphenols, flavonoids and terpenoids compounds, and in their antioxidant activity (Figure 2; Table 2). Of note, these three classes of compounds have anti-inflammatory, antibacterial and antiviral activities [37], and therapeutic effects against cancer [38,39], cardiovascular [40] and neurodegenerative diseases [41]. They act on different pathways involved in cancer and are able to inhibit DNA damage and oxidative stress induced by free radicals [1,42,43,44]. In this work, polyphenols and terpenoids have been found in all fractions except for the sub-fraction I that contained only polyphenols (Figure 2). Remarkably, we observed an enrichment of Total Terpenoid Content (TTeC) and a variable Total Polyphenol Content (TPC) passing from II to VIII sub-fractions. II and IX sub-fractions also showed flavonoids, a group of natural substances with variable phenolic structures that are well known for their beneficial effects on health [45]. The free radical-scavenging potential of flavonoids has a great importance in protecting the body against reactive oxygen species, such as superoxide radical, hydrogen peroxide, singlet oxygen and hydroxyl radical, that are implicated in the cause or progression of several human diseases, including cancer [46,47]. Importantly, according to these considerations, we found that the only sub-fractions containing flavonoid compounds (II and IX) showed the highest SO scavenging activity followed by VIII, X and V sub-fractions (Table 2). Our results showed as the antioxidant capacity is directly proportional to the highest content of polyphenols; in fact X, II, IX and V sub-fractions showed the highest RACI values and had the high polyphenols content.

We found that the majority of I-X sub-fractions induced a reduction of KG1 and MV4-11 cell viability in a dose and time dependent way (Figure 3). These results are in accordance with previous studies reporting the cytotoxic effects of different *Azorella* species on a variety of solid and hematological malignancies [34,36], including AML [14]. Moreover, our results indicated that differences in the cytotoxic activities among I-X sub-fractions seem to be inversely associated with their in vitro antioxidant activities. In particular, IV and VI sub-fractions resulted cytotoxic against cancer cells, while they did not show antioxidant activity on the basis of the RACI index. Therefore, we believe that the cytotoxic effect of IV and VI sub-fractions could be associated with the activation of other death signaling pathways including Notch, WNT/β-catenin, Hedgehog and PI3/AKT [48]. IV and VI sub-fractions induced a greater reduction of AML cell vitality compared to control already at 24 h of treatment with the lowest concentration (10 μg/mL). These data also emphasized as the fractionation process by column chromatography gave us sub-fractions enriched in bioactive compounds. Another previous study reports the use of this fractionation technique for a primary enrichment of compounds present in a mixture of substances [49].

Further confirmation of the enrichment in bioactive compounds in IV and VI sub-fractions was provided by the comparison of their average EC_50_ value calculated on KG1 and MV4-11 cells at 48 h of treatment with that of AG CHCl_3_ at 48 h. In particular, it was halved, passing from 43 μg/mL in AG CHCl_3_ to 20 μg/mL in both sub-fractions at 48 h with an increase in activity (Table 1, Table 2 and Table 3). Notably, recent work reported that natural extracts having an EC_50_ of about 20 μg/mL could be investigated as potential anticancer agents [50]. 

In addition, IV and VI sub-fractions, showing a negligible reduction of healthy cells viability (Figure 3c,d), may become promising “ideal” anticancer agents. In fact, it is well known that many plant extracts are safer and induce few side effects than synthetic drugs, making them interesting as alternative and complementary agents in cancer [51]. All these data encourage further examination for possible therapeutic application of IV and VI sub-fractions in AML.

Anticancer activity is usually mediated by the inhibition of proliferation, the induction of apoptosis and the arrest of cell cycle [52]. Apoptosis is a physiological cell death occurring through intrinsic and extrinsic interconnected pathways [53,54]. These mechanisms are usually deregulated in cancer, leading to malignant transformation and resistance to anticancer drugs [55]. Interestingly, IV and VI sub-fractions increased apoptosis in both AML cell lines (Figure 4). The apoptosis activation was confirmed by PARP-1 involvement, whose cleavage is considered a central indicator of this pathway [56]. In particular, IV and VI sub-fractions specifically activated, in both AML cell lines, the intrinsic apoptotic pathway as evidenced by the decrease of the apoptosis inhibitor protein Bcl2 (Figure 5) [57]. The activation of this pathway is one of the key mechanisms involved in the role of many antitumor drugs [58], so its identification provides indications for new possible target therapies. Our results are in accordance with other recent works reporting the activation of the pathway after treatment with different plant extracts [59,60]. Interestingly, we confirmed the increase of apoptosis after treatment with IV and VI also in primary cells of three AML patients with diverse mutational status (Figure 7). In particular, one showed nucleophosmin 1 (NPM1) gene mutated, and two carried both NMP1 and the internal tandem duplication (ITD) mutation in the FMS-like tyrosine kinase 3 (FLT3) gene. Patients with mutated NPM1 with no or low FLT3-ITD have a favorable risk status and a good response to chemotherapy [61].

Additionally, cell cycle deregulation is another hallmark in tumor cells and mutations in its checkpoint genes are frequent in cancer [52]. According with it, our results showed that IV and VI sub-fractions increased the percentage of cells in G_0_/G_1_ phase and reduced those in G_2_/M phase in both AML cell lines (Figure 6).

We hypothesized that the antitumor activity of IV and VI sub-fractions could be due to the presence of common compounds in both sub-fractions that could target several pathways in AML subtypes. In fact, it is well known that natural mixture, like plant extracts, contain various types of secondary metabolites that, for their heterogeneity in chemical structures, are able to target multiple tumor pathways [62]. In agreement to us, recent studies report both the in vitro and in vivo anti-leukemic properties effects of other plant extracts [28,63]. Although treatment with IV and VI sub-fractions induced the reduction of cell viability, the increase of apoptosis and the block of the cell cycle in both KG1 and MV4-11 cells, the reduction of viability and cell cycle arrest were more pronounced on MV4-11 compared to KG1 (Figure 3 and Figure 6). This could be due to the different genetic background of KG1 and MV4-11 cell lines and, consequently, to their cellular context.

Mass spectrometry analysis confirmed the presence of compounds common to IV and VI sub-fractions that were tentatively identified as mulinic acid or azorellane and their derivatives (Table 4 and Table 5). They are terpenoid compounds already isolated from *A. compacta* Phil. and other species belonging to the Apiaceae family [64]. Many studies have been reported the different biological activities of these terpenoid compounds, including their antibacterial [31], antituberculosis [65], gastroprotective [30] and antitumor activity [34,36]. In particular, many of the putatively compounds present in AG have important cytotoxic activity; for examples, Bórquez et al. showed cytotoxic effect of azorellanes on human breast carcinoma cells; their activity was dependent on their beta acetate group at position *C*-7 in the skeleton [34]. Moreover, different studies reported the cytotoxic properties of *Azorella* extracts on different solid and hematological malignancies [14,34,36]. However, the information about the composition of IV and VI sub-fractions are putative for the lack of available standards, so additional investigations are needed to define the precise chemical compositions of the above sub-fractions and to allow the isolation and fine structural characterization of pure compounds.

In conclusion, our data candidate *Azorella glabra* plant as a possible source of new natural agents against AML cells. Specifically, two AG chloroform sub-fractions showed strong cytotoxic effect against AML cells. Mass spectrometry characterization suggests the presence of specific terpenoids, like mulinic acid or azorellane and their derivatives, in these sub-fractions. However, further studies should be conducted to isolate bioactive compounds, to test their action alone and/or in combination with current pharmacological treatments, and to investigate the molecular mechanisms underlying anti-AML activity.

## 4. Materials and Methods

### 4.1. Chemicals, Reagents and Instruments

Solvents as chloroform (CAS Number 67-66-3), *n*-hexane (CAS Number 110-54-3), ethanol (CAS Number 64-17-5), and methanol (CAS Number 67-56-1) were purchased from Carlo Erba (Milan-Italy). Acetonitrile (CAS Number 75-05-8) and formic acid (CAS Number 64-18-6) were purchased from Merck (Wicklow-Ireland). Folin-Ciocalteu reagent 2N (MDL number MFCD00132625), sodium carbonate (CAS Number 497-19-8), 2,2-azino-bis(3-ethylbenzothiazoline-6-sulfonic acid) (ABTS, MDL number MFCD00214141), potassium persulfate (CAS Number 7727-21-1), potassium phosphate monobasic (CAS Number 7778-77-0), *β*-nicotinamide adenine dinucleotide reduced form (NADH, CAS Number 104809-32-7), phenazinemethosulfate (PMS, CAS Number 299-11-6), nitrotetrazolium blue chloride (NBT, CAS Number 298-83-9), sodium nitroprusside dehydrate (SNP, CAS Number 13755-38-9), sulfanilamide (CAS Number 63-74-1), *N*-(1-Naphthyl) ethylenediaminedihydrochloride (CAS Number 1465-25-4), sodium acetate trihydrate (CAS Number 6131-90-4), anti-rabbit secondary antibody, dimethyl sulfoxide (DMSO, CAS Number 67-68-5), and Ficoll Histopaque-1077 were purchased from Sigma-Aldrich (Milano, Italy). Standards as 6-hydroxy-2,5,7,8-tetramethylchroman-2-carboxylic acid (Trolox, CAS Number 53188-07-1), ascorbic acid (CAS Number 50-81-7), gallic acid (CAS Number 149-91-7), and Leucine-Enkephalin (MDL number MFCD11045938) were purchased from Sigma-Aldrich and Merck, respectively. Water was deionized using a Milli-Q water purification system (Millipore, Bedford, MA, USA). All spectrophotometric measurements were done in 96-well microplates or cuvettes on an UV/Vis spectrophotometer (SPECTROstar^Nano^ BMG Labtech, Ortenberg-Germany), and each reaction was performed in triplicate. LC–MS/MS analysis was performed on a Q-Tof Premier mass spectrometer (Waters Corporation, Milford, MA, USA) coupled to an Alliance 2695 HPLC system (Waters Corporation). Fetal bovine serum (FBS), Roswell Park Memorial Institute 1640 Medium (RPMI-1640), Dulbecco’s Modified Eagle Medium (DMEM), phosphate-buffered saline (PBS) and penicillin/streptomycin were purchased from Gibco-BRL (Life technologies, Carlsbad, CA, USA). The cellTiter 96 Aqueous One Solution assay kit (tetrazolium compound [3-(4,5-dimethylthiazol-2-yl)-5-(3-carboxymethoxyphenyl)-2-(4 sulfophenyl)-2*H* -tetrazolium, inner salt; MTS) was purchased from Promega (Madison, WI, USA). The fluorescein isothiocyanate (FITC) Annexin V Apoptosis Detection kit was purchased from Becton Dickinson (BD Pharmingen, San Jose, CA, USA). The Intracell kit was purchased from Immunostep (Salamanca, Spain). Antibody anti-PARP-1 was purchased from CST (Danvers, MA, USA). Antibody anti-Bcl-2 and chemiluminescence (ECL plus) kit were purchased from Thermo Fisher (Rodano, Milano, Italy). Nitrocellulose membrane was purchased from GE Healthcare (Chicago, IL, USA).

### 4.2. Chromatographic Separation of AG CHCl_3_ Fraction

The aerial parts of AG, grown in natural conditions, were collected in 2014 near the Aymaya population/community (18.45° S to 66.46° W; 3750 msnm), Bustillo province, Potosí department, Bolivia, and a voucher specimen was stored at the University of La Paz. As previously reported, in the herbal medicinal plants of the National University Siglo XX, Llallagua, Potosí, Bolivia, are found the samples of the species [2]. The ethanol extract of dried AG aerial parts was subjected to liquid/liquid extraction, giving AG CHCl_3_ fraction [2]. Then, 2.5 g of AG CHCl_3_ fraction have been solubilized in chloroform:*n*-hexane 9:1 and were subjected to a further separation by silica gel column chromatography (100 cm × 1 cm, silica 60–200 µm) and eluted with a mixture of *n*-hexane, chloroform (CHCl_3_), and methanol (MeOH) with an increasing polarity. From silica gel column chromatography, 1043 fractions were obtained, and all of them were analyzed by Thin Layer Chromatography (TLC) in order to compare the chromatographic profile and re-unite the simile profiles. TLC was performed on silica gel plates F254 on a glass support; as mobile phases, different solvent mixtures were selected between *n*-hexane, CHCl_3_, and MeOH. Then, fractions with similar chromatographic pattern were combined, and, at the end, we had 10 different sub-fractions (I-X).

### 4.3. Determination of Total Polyphenol Content (TPC), Total Flavonoid Content (TFC), and Total Terpenoid Content (TTeC)

The AG CHCl_3_ fraction and its 10 sub-fractions (I-X) were analyzed to determine their TPC, TFC, and TTeC using the method previously reported [2]. In detail, the Folin-Ciocalteu reagent was added to different concentrations of samples in order to determine the TPC content of samples. The gallic acid was used as reference standard and the obtained data were expressed as mg of Gallic Acid Equivalents per gram of dried sample (mg GAE/g). 

Moreover, a mixture of AlCl_3_, NaNO_3_, and NaOH was used to determine the TFC of samples expressing the results in mg of Quercetin Equivalents per gram of dried sample (mg QE/g).

Then, a rapid and high-throughput assay was employed to determine the total terpenoids content in samples. The regression equation of the monoterpene Linalool standard curve was used to express the results (mg of Linalool Equivalents per gram of dried sample or mg LE/g).

### 4.4. Determination of Antioxidant Activities

All AG samples were tested for their antioxidant activity using the synthetic ABTS^·+^ radical and the biological superoxide anion (SO assay) and nitric oxide (NO assay) radicals in three different in vitro assays, as previously reported [2]. The Trolox was used as reference standard in the reaction test with ABTS^·+^ radical generated after 16 h of incubation between ABTS salt and potassium persulfate. The results obtained at 734 nm were expressed as milligrams of Trolox Equivalents per gram of dried sample (mg TE/g).

The NADH/PMS in vitro system and nitroprusside were used to generate superoxide anion and nitric oxide radicals, respectively, in two different in vitro assays. 

The scavenging activity of tested samples against superoxide anion radicals was monitored as the inhibition of formazan formation at 560 nm. Instead, the Griess reagent was used to determine the interaction between nitric oxide and oxygen. In both assays, the ascorbic acid was used as positive control, and results were expressed as the concentration inhibiting 25% of radical inhibition in μg/mL (IC_25_).

Moreover, RACI index has been calculated using the results coming from different antioxidant chemical methods (ABTS and SO assays) together with TPC in Excel software (Microsoft, Washington DC, WA, USA) in order to rank the antioxidant capacity derived from different antioxidant methods. In particular, this adimensional index score was calculated on the mean value and standard deviation of all assays, subtracting the mean from the raw data divided by the standard deviation.

### 4.5. AG samples Preparation

All AG samples were dried by rotary evaporator (IKA RV 10) and stored in dark at 4 °C until use; just before the use, the samples were dissolved in DMSO at the stock solution of 30 mg/mL and then diluted in RPMI-1640 medium for cell treatments.

### 4.6. Cell Lines, Healthy Donors and AML Patients

An erythroleukemia FLT3 wild type cell line, KG1, and a monocytic leukemia cell line carrying FLT3-ITD mutation, MV4-11, were cultured in RPMI-1640 supplemented with 10% FBS, 1% of penicillin–streptomycin. After obtaining informed consent, PB from five healthy subjects and BM blood from three AML patients were drawn into EDTA tubes. All experiments on cells were conducted followed guidelines and regulations of the Helsinki Declaration and were approved by the Ethics Committee of IRCCS CROB (Prot 3 725; 7-2-2008). Patient characteristics are shown in Table 6. PBMCs and BMMCs were obtained by Ficoll Hystopaque gradient separation.

PBMCs were cultured in DMEM supplemented with 20% FBS, 1% of penicillin–streptomycin. BMMCs were cryopreserved in FBS containing 10% DMSO, stored at −196 °C in liquid nitrogen storage tank, and, before treatment, they were appropriately thawed and cultured in DMEM supplemented with 20% FBS, 1% of penicillin–streptomycin at 37 °C for 2 h. All cells were grown at 37 °C in 5% CO_2_.

### 4.7. Cell Viability Assay

Cell viability was assessed using MTS assay. In brief, AML cell lines and PBMCs were seeded into 96-well plates (3 × 10^4^ cell/100 μL medium), treated with AG CHCl_3_ and I-X sub-fractions at different common concentrations (10, 25, and 50 μg/mL), and incubated at 37 °C for 24 and 48 h. The absorbance was measured at 492 nm using VICTOR Nivo (Perkin Elmer, Waltham, MA, USA). The optical density (OD) of DMSO vehicle treated cells was considered as 100% of viability. All experiments were conducted in triplicate. Cell viability was expressed as percentage of AG treated viable cells vs DMSO vehicle treated cells using the following formula:Percentage of cell viability = (AG treated cell OD)/ (DMSO treated cell OD) × 100.(1)

The concentration of AG samples able to inhibit AML cell growth of 50% (EC_50_) was determined using GraphPad Prism (GraphPad Prism, San Diego, CA, USA) [2].

### 4.8. Apoptosis Assay

KG1 and MV4-11 cell lines and primary AML cells were seeded in a 6-well culture plate at a density of 4 × 10^5^ cell/1300 μL medium. For the treatment, we used different concentrations of IV and VI sub-fractions for the two AML cell lines because we wanted to evaluate antitumor effects on cells with adverse risk status, like MV4-11 cells, using low concentration of both sub-fractions. Moreover, the lower EC_50_ value calculated on MV4-11 cells at 48 h compared to that calculated on KG1 cells at the same time supported our choice. Based on these data, KG1 and primary AML cells were treated with 30 μg/mL of IV and VI sub-fractions; MV4-11 cells were treated with 10 μg/mL of the same sub-fractions. AML cells were treated for 24 and 48 h.

Apoptotic cells were detected by cytometric and western blot analysis, as previously reported [2].

For the cytometric assay, a FITC Annexin V Apoptosis Detection kit was used, following the manufacturer’s protocol. In particular, after treatment, AML cells were harvested, washed and resuspended in Annexin V binding buffer. Next, cells were labeled with 5 μL of FITC Annexin V and 5 μL of propidium iodide (PI) and were incubated at dark for 15 min. A total of 1 × 10^4^ events were acquired using NAVIOS flow cytometer and analyzed by Kaluza 2.0 software (Beckman Coulter, Life Sciences, Indianapolis, IN, USA). Both single positive for Annexin V and double positive for Annexin V and PI cells were interpreted as signs of early and late phases of apoptosis, respectively.

For western blot analysis, a total of 2 × 10^6^ AML cells were collected after treatment, lysed in buffer (50 mM Tris pH 7.4, 150 mM NaCl, 1% NP40, 0.1% SDS, 0.5% sodium deoxycholate) supplemented with protease inhibitors cocktail (Sigma Aldrich, Milano-Italy) and centrifuged at 13,000× *g* for 30 min at 4 °C. The protein concentration in each sample was detected using the Bradford assay. Eighty micrograms of sample lysate were separated by SDS-PAGE (4–8% for PARP and 4–15% for Bcl2) and transferred to nitrocellulose membrane. Nonspecific binding sites were blocked with TBST buffer containing 5% nonfat dry milk (NFM) at room temperature for 1 h. Subsequently, membranes were incubated with primary antibodies anti-PARP-1 (1:1000 in 5% NFM/TBST) and anti-Bcl-2 (1:200 in 5% NFM/TBST) overnight at 4 °C. After incubation with HRP-conjugated anti-rabbit secondary antibody (1:10000), detection was performed using the enhanced chemiluminescence (ECL plus) kit. Densitometric analysis was performed using IMAGEJ software, downloaded from the NIH website (http://imagej.nih.gov/ij/).

### 4.9. Cell Cycle Analysis

KG1, MV4-11, and primary AML cells were treated as described in the previous section. After treatment, KG1 and MV4-11 cells were harvested, washed and fixed in cold ethanol 70% for 1 h. Instead, AML primary cells were harvested, washed, fixed, and permeabilized by Intracell kit, as previously described [66]. Fixed cells were then labeled with PI/RNase staining solution for 30 min at room temperature at dark. A total of 1 × 10^4^ events were acquired by NAVIOS flow cytometer and analyzed by Kaluza 2.0 software (Beckman Coulter, Life Sciences, Indianapolis, IN, USA).

### 4.10. LC-MS/MS Characterization of Samples

The identification of the compounds in IV and VI sub-fractions was obtained by liquid chromatography mass spectrometry (LC-MS/MS) on a Q-Tof Premier mass spectrometer (Waters Corporation, Milford, MA, USA) coupled to an Alliance 2695 HPLC system (Waters Corporation, Milford, MA, USA) using the same parameters of a validated method previously described and employed for *A. glabra* samples [10]. In particular, a negative ionization mode was used to acquire the electrospray mass spectra data for a mass range *m/z* 100 to *m/z* 1000 in according to the present literature [67,68]. An Atlantis T3 C18 column was used (Waters Corporation, Milford, USA, 100.00 × 2.10 mm; 3.00 µm particle size) at 40 °C. The mobile phases were 0.10% aqueous formic acid (solvent A) and 0.10% formic acid in acetonitrile (solvent B). The stepwise gradient from 10% to 90% solvent B was applied at flow rate of 300 µL/min for 25 min. Cone voltage and capillary voltage were set at 30 V and 3 kV, respectively. Argon was used as collision gas and the collision induced fragmentation (CID) of the analytes was achieved using 12 to 30 eV energy. The injection volume for all the samples was 3.00 µL. 

### 4.11. Statistical Analysis

All data were expressed as mean ±standard deviation (SD) of three independent experiments. The *p* values were analyzed by one-way analysis of variance (ANOVA) (for the determination of TPC, TFC, TTeC, for the comparison of the cellular viability of all sub-fractions respect to AG CHCl_3_ fraction and for the western blot analysis of apoptosis) and unpaired *t*-test (for the viability assay of AG CHCl_3_ fraction on KG1 and MV4-11 cells, for the viability assay on healthy PBMCs and for the cytofluorimetric evaluation of apoptosis and cell cycle) determined using GraphPad Prism 5 Software (San Diego, CA, USA).

## Figures and Tables

**Figure 1 molecules-25-04890-f001:**
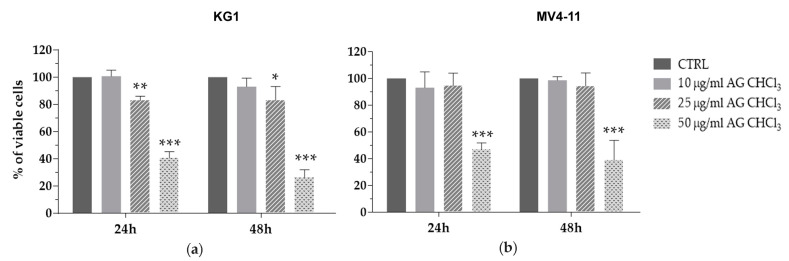
Cell viability in acute myeloid leukemia (AML) cell lines after treatment with *Azorella glabra* (AG) chloroform fraction (AG CHCl_3_). KG1 (**a**) and MV4-11 (**b**) cells were treated with AG CHCl_3_ fraction at different concentrations (10, 25, and 50 μg/mL) for 24 h and 48 h. Results are expressed as percent of cell viability normalized to dimethyl sulfoxide (DMSO)-treated control cells (CTRL). Bar-graphs represent mean ± standard deviation (SD) from three independent experiments. Statistically significant analyses from unpaired *t*-test are indicated by asterisks: * *p* ≤ 0.05, ** *p* ≤ 0.01, *** *p* ≤ 0.001.

**Figure 2 molecules-25-04890-f002:**
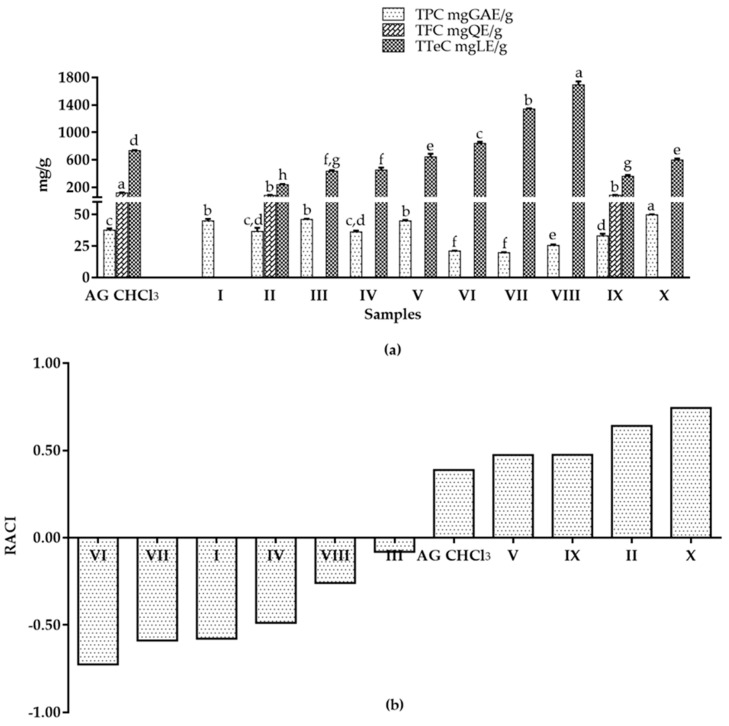
Phytochemical profile and in vitro antioxidant activity of AG CHCl_3_ and its sub-fractions. (**a**) Total Polyphenol Content (TPC), Total Flavonoid Content (TFC), and Total Terpenoid Content (TTeC) of AG samples. Results were expressed as mean ± SD of triplicate determinations in mg of Gallic Acid Equivalents per gram of dried sample (mg GAE/g), in mg of Quercetin Equivalents per gram of dried sample (mg QE/g), and in mg of Linalool Equivalents per gram of dried sample (mg LE/g). For each test, values with identical letters (a–g) are not significantly different at the *p* < 0.05 level, 95% confidence limit, according to one-way analysis of variance (ANOVA). (**b**) Relative Antioxidant Capacity Index (RACI) of AG samples.

**Figure 3 molecules-25-04890-f003:**
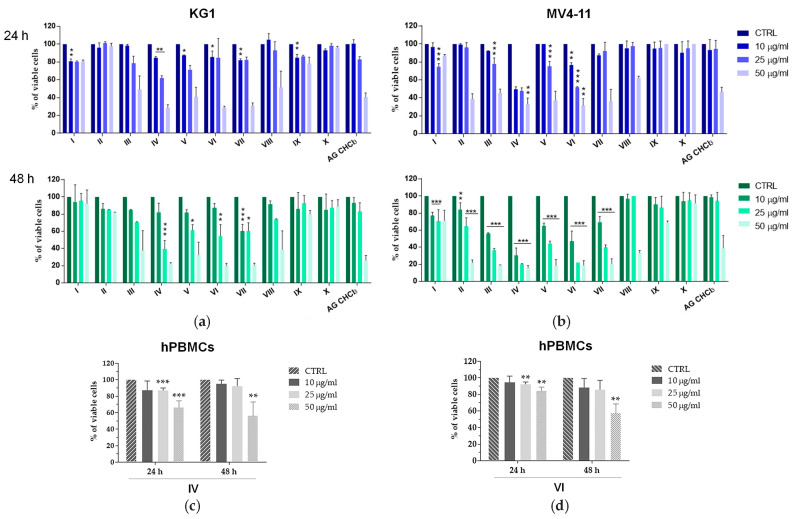
Cell viability in AML cell lines and healthy peripheral blood mononuclear cells (hPBMCs) after treatment with AG samples. KG1 (**a**), MV4-11 (**b**), and five hPBMCs (**c**), (**d**) were treated with AG samples at different concentrations (10, 25, and 50 μg/mL) for 24 h and 48 h. Results are expressed as percent of cell viability normalized to DMSO-treated control cells. Bar-graphs represent mean ± SD from three independent experiments. Statistically significant analyses from two-way ANOVA comparing all sub-fractions (I-X) respect to AG CHCl_3_ fraction (**a**,**b**) and from *t*-test (**c**,**d**) are indicated by asterisks: * *p* ≤ 0.05, ** *p* ≤ 0.01, *** *p* ≤ 0.001.

**Figure 4 molecules-25-04890-f004:**
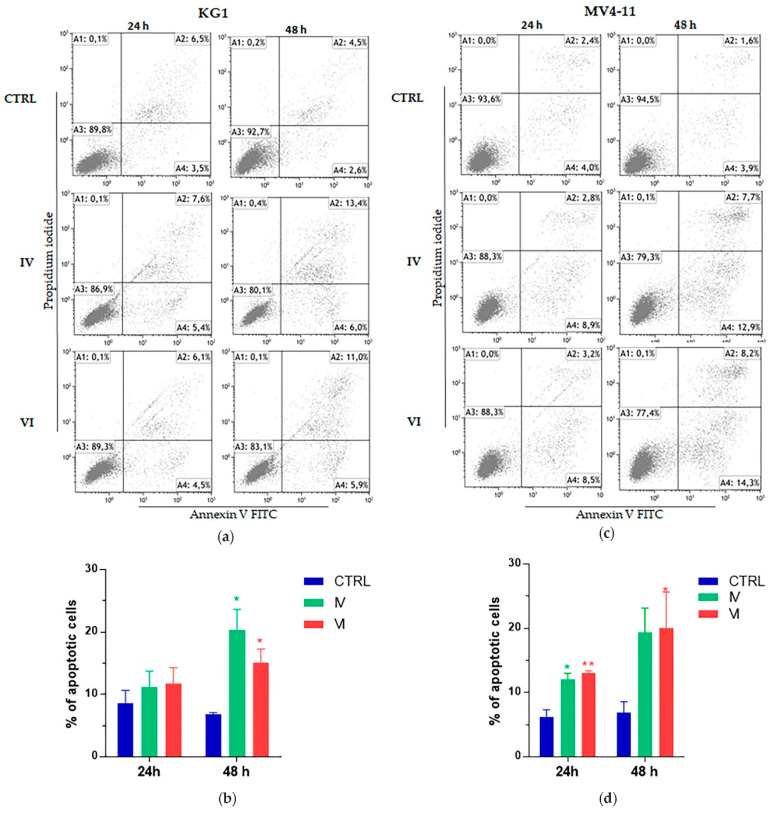
Cytofluorimetric evaluation of apoptosis on AML cell lines after treatment. KG1 and MV4-11 cell lines were treated with 30 μg/mL and 10 μg/mL, respectively, of IV and VI sub-fractions for 24 h and 48 h. Dot plots show a representative experiment on KG1 (**a**) and MV4-11 (**c**) cells after 24 h and 48 h of treatment; bar-graphs represent mean ± SD from three independent experiments on KG1 (**b**) and MV4-11 (**d**) cells, considering both early and late apoptosis. Statistically significant analyses from unpaired t-test are indicated by asterisks: * *p* ≤ 0.05, ** *p* ≤ 0.01.

**Figure 5 molecules-25-04890-f005:**
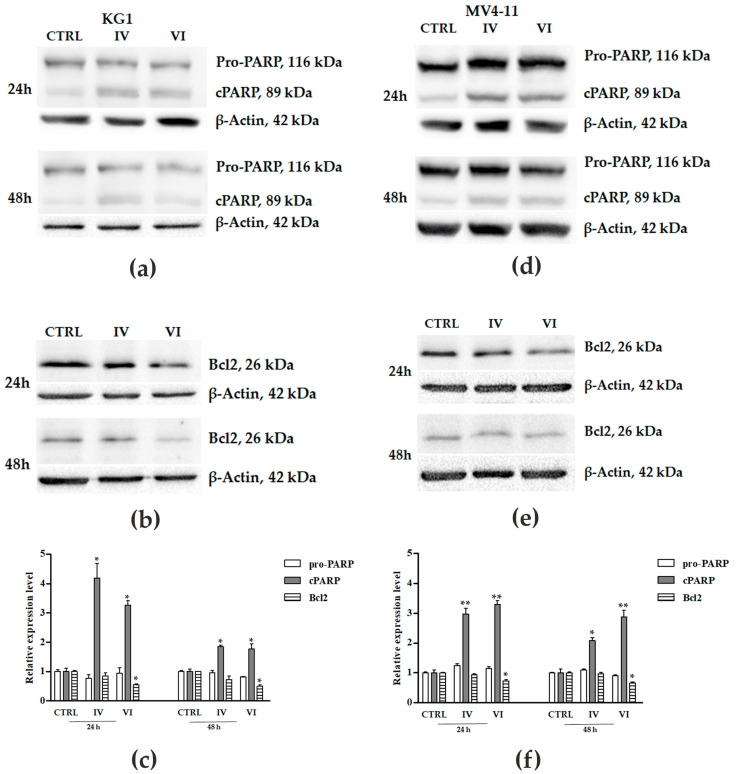
Western blot analysis of apoptosis-related proteins on AML cell lines after treatment. Expression of pro-PARP-1, cleaved PARP (cPARP) and Bcl2 was evaluated in KG1 (**a**,**b**) and MV4-11 (**d**,**e**) cells treated and not (vehicle-treated cells, CTRL) with 30 μg/mL (for KG1) and 10 μg/mL (for MV4-11) of IV and VI sub-fractions for 24 h and 48 h. *β*-actin was used as a loading control. Figures and bar-graphs are representative of three independent experiments in KG1 (**c**) and MV4-11 (**f**) cells. Data were expressed as means ± SD. Statistically significant analyses from one-way ANOVA followed by Tukey’s Multiple Comparison test are indicated by asterisks: * *p* ≤ 0.05, ** *p* ≤ 0.01.

**Figure 6 molecules-25-04890-f006:**
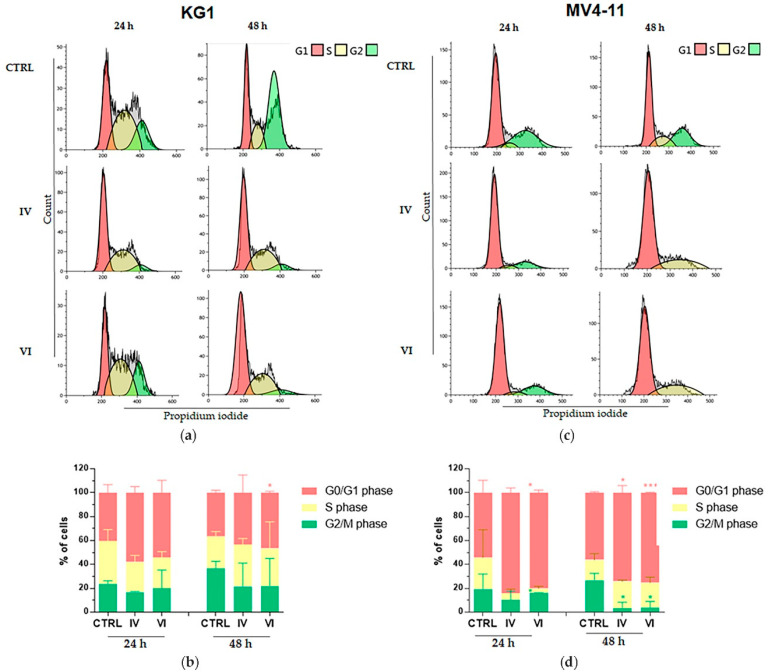
Cytofluorimetric evaluation of AML cells distribution in the cell cycle after treatment. Cell cycle histograms show a representative experiment on KG1 (**a**) and MV4-11 (**c**) cells treated with 30 μg/mL (for KG1) and 10 μg/mL (for MV4-11) of IV and VI sub-fractions for 24 h and 48 h compared to untreated control cells (CTRL); the bar-graphs represent mean of three independent experiments on KG1 (**b**) and MV4-11 (**d**) cells with SD. Statistically significant analyses from unpaired *t*-test are indicated by asterisks: * *p* ≤ 0.05, *** *p* ≤ 0.001.

**Figure 7 molecules-25-04890-f007:**
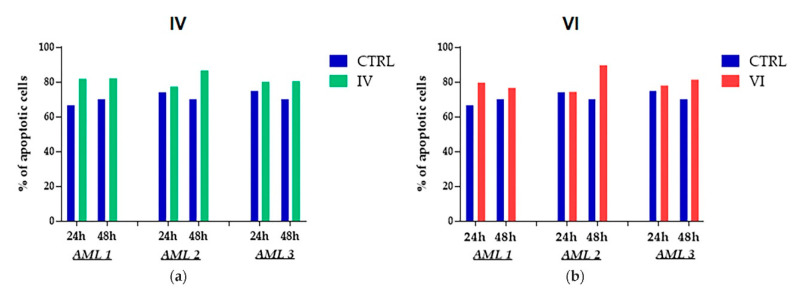
Cytofluorimetric evaluation of apoptosis on primary AML cells after treatment. Bone marrow mononuclear cells (BMMCs) derived from three AML patients were treated with 30 μg/mL of IV (**a**) and VI (**b**) sub-fractions for 24 h and 48 h. Percent of apoptotic cells was obtained from the sum of early and late apoptosis of a single experiment.

**Figure 8 molecules-25-04890-f008:**
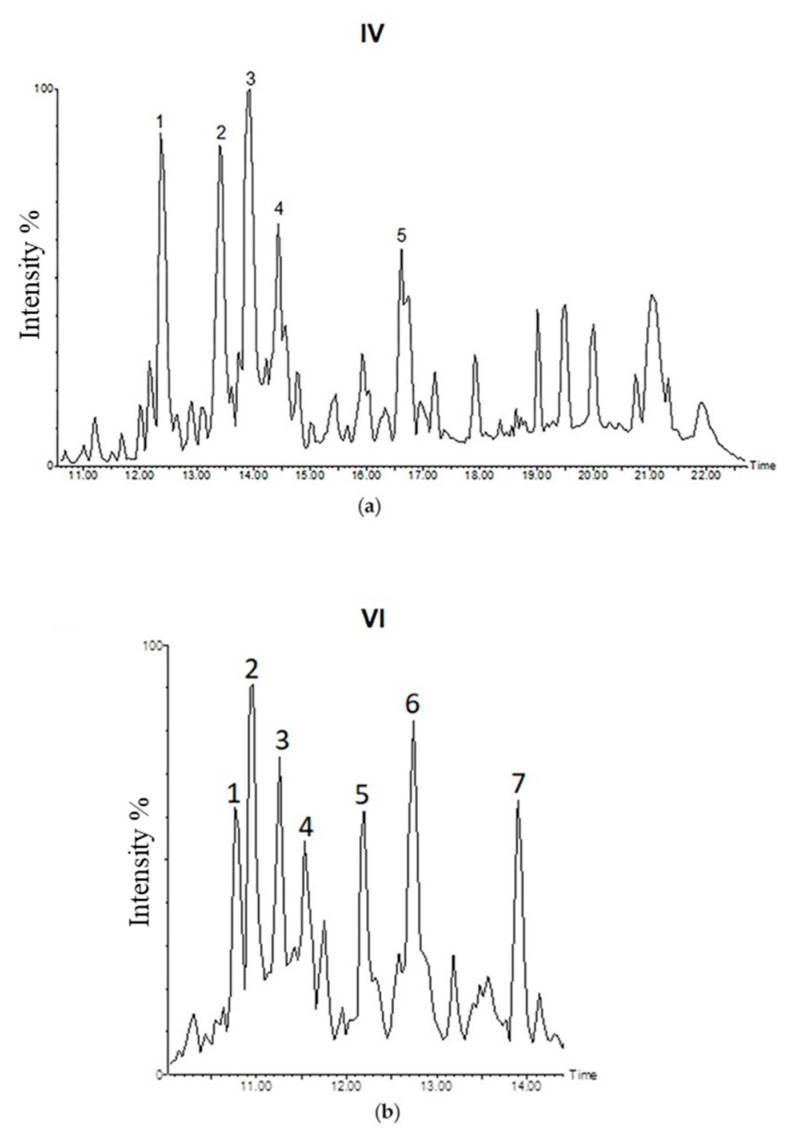
LC-MS/MS analysis of AG sub-fractions. IV (**a**) and VI (**b**) sub-fractions base peak intensity chromatogram.

**Table 1 molecules-25-04890-t001:** EC_50_ values of the AG CHCl_3_ fraction on AML cells.

Cell Lines	EC_50_ (μg/mL)	
	24 h	48 h
KG1	45.42	39.90
MV4-11	49.00	46.53

**Table 2 molecules-25-04890-t002:** 2,2-azino-bis(3-ethylbenzothiazoline-6-sulfonic acid) (ABTS) and Super Oxide (SO) Scavenging Activity of AG samples.

Samples	ABTS (mgTE/g)	SO (IC_25_ μg/mL)
AG CHCl_3_	32.08 ± 0.02 ^a^	470.00 ± 20.00 ^c^
I	nd	188.72 ± 16.68 ^b^
II	22.85 ± 0.57 ^b^	46.31 ± 2.05 ^a,b^
III	15.14 ± 0.64 ^c^	1996.03 ± 97.23 ^e^
IV	13.08 ± 0.22 ^d^	2171.71 ± 125.59 ^f^
V	17.86 ± 1.05 ^e^	67.78 ± 1.80 ^a,b^
VI	12.48 ± 0.43 ^d^	115.41 ± 0.85 ^a,b^
VII	23.25 ± 0.72 ^b^	1174.48 ± 43.23 ^d^
VIII	11.65 ± 0.55 ^d^	52.08 ± 2.38 ^a,b^
IX	16.40 ± 0.50 ^c,e^	35.96 ± 3.04 ^a^
X	17.70 ± 0.64 ^e^	54.80 ± 4.47 ^a,b^

Results were expressed as mean ± standard deviation of triplicate determinations in mg of Trolox Equivalents per gram of dried sample (mg TE/g) and in concentration of the sample required to inhibit the activity of the radical by 25% (IC_25_ μg/mL). For each test, the values with identical letters (a–f) are not significantly different at the *p* < 0.05 level, 95% confidence limit, according to one-way analysis of variance (ANOVA). Samples are chloroform fraction (AG CHCl_3_) and its sub-fractions (I-X); nd = not determined at tested concentrations.

**Table 3 molecules-25-04890-t003:** EC_50_ values of the sub-fraction IV and VI on AML cells.

Fraction	Cell Lines	EC_50_ (μg/mL)	
		24 h	48 h
IV	KG1	35.40	27.60
	MV4-11	28.50	8.99
VI	KG1	33.72	30.40
	MV4-11	32.91	9.87

**Table 4 molecules-25-04890-t004:** Liquid chromatography-tandem mass spectrometry (LC–MS/MS) of AG IV sub-fraction.

Peak No.	Retention Time (min)	ESI (-) MS Observed	ESI (-) MS Calc.	Molecular Formula	Δppm	MS/MS	Tentative Identity
1	12.36	333.2072	333.2071	C_20_H_30_O_4_	0.3	289, 287, 271, 253, 229, 135, 113, 95, 87, 83, 69, 57	Mulinic acid or 14*α*-hydroxymulin-12-en-11-one-20-oic acid [25]
2	13.40	333.2072	333.2071	C_20_H_30_O_4_	0.3	289, 287, 271, 253, 229, 135, 113, 95, 87, 83, 69, 57	Mulinic acid or 14*α*-hydroxymulin-12-en-11-one-20-oic acid [25]
3	13.94	331.1917	331.1915	C_20_H_28_O_4_	0.6	287, 269, 259, 243, 229, 215, 121, 111, 109, 96, 83	Mulin-12-en-11, 14-dion-20-oic acid [26]
4	14.43	303.1945	303.2345	C_20_H_32_O_2_	4.9	259, 249, 243, 217, 189, 135, 123, 95, 81, 57	Azorellanone 33 or Dihydroazorellolide [28]
5	16.61	497.2734	497.2756	C_26_H_42_O_9_	−4.4	333, 271, 229, 135, 113, 95, 83, 69, 57	Mulinic acid hexoside derivative

**Table 5 molecules-25-04890-t005:** Liquid chromatography-tandem mass spectrometry (LC–MS/MS) of AG VI sub-fraction.

Peak No.	Retention Time (min)	ESI (-) MS Observed	ESI (-) MS Calc.	Molecular Formula	Δppm	MS/MS	Tentative Identity
1	10.76	335.2238	335.2230	C_20_H_32_O_4_	2.4	289, 273, 245, 235, 59	13,14-dihydroxymulin-11-en-20-oic acid [29]
2	10.97	335.2238	335.2230	C_20_H_32_O_4_	2.4	289, 273, 245, 235, 59	13,14-dihydroxymulin-11-en-20-oic acid [29]
3	11.26	349.2368	349.2384	C_21_H_34_O_4_	−4.6	331, 305, 287, 269, 259, 245, 235, 219, 205, 193, 177, 151, 123, 111, 99, 83, 69, 59, 57	Mulin-en-dion-oicacid derivative
4	11.53	335.2238	335.2230	C_20_H_32_O_4_	2.4	289, 273, 245, 235, 59	13,14-dihydroxymulin-11-en-20-oic acid [29]
5	12.19	393.2427	393.2430	C_26_H_34_O_3_	−0.8	353, 335, 289, 273, 245, 235, 59	Dihydroxymulinic acid derivative
6	12.74	333.2072	333.2071	C_20_H_30_O_4_	0.3	289, 287, 271, 253, 229, 135, 113, 95, 87, 83, 69, 57	Mulinic acid or 14*α*-hydroxymulin-12-en-11-one-20-oic acid [25]
7	13.90	331.1917	331.1915	C_20_H_28_O_4_	0.6	287, 269, 259, 243, 229, 215, 121, 111, 109, 96, 83	Mulin-12-en-11, 14-dion-20-oic acid [26]

**Table 6 molecules-25-04890-t006:** AML patient characteristics.

AML Patients	Age	Sex	Mutation	Blast (%)
1	68	Female	FLT3-ITD NPM1	>90
2	78	Female	NPM1	60–70
3	58	Female	FLT3-ITD NPM1	100

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
