# Peer review of "Advances in Azorella glabra Wedd. Extract Research: In Vitro Antioxidant Activity, Antiproliferative Effects on Acute Myeloid Leukemia Cells and Bioactive Compound Characterization"

_molecules, 2020, doi:10.3390/molecules25214890_

Round 1

Reviewer 1 Report

  The authors show the anti-AML effect of Azorela glabra Wedd. chloroform fraction and its ten sub-fractions (I-X) in the present manuscript. The further investigation may be utilized in a clinical setting.  

Major comments.

  1. Although sub-fractions other than IV and VI also showed decresed cell viability, the authors performed futher analysis (apoptosis assay) using only IV and VI. Can you show the data using other sub fractions? 
  2.  The authors speculated the possible role of mulicinic acid, or azorellane and their derivatives in anti-AML activity by LC-MS/MS analysis.  Have you analyzed the possible molecules in sub-fractions  other than IV and VI? 
  3.  In Fig 7, % apoptotic cells on 24 h in AML3 seem to be marginal.
  4.  The authors described AML petients with NPM1 with no or low FLT3-ITD mutation have a favorable prognosis in Discussion. Is this general information concordant with the findings in Fig 7.

Minor comment.

1. In line 46, "ad" should be "and".

Author Response

Pleese see the attachment.

Reviewer 2 Report

The authors performed an exploration of the effect of chloroformic extract of Azorella glabra Wedd continuing their previous research. They go deeper and explore their content in terms of polyphenols, flavonoids, terpenoids. Additionally, the antioxidant properties are explored for all sub-fractions and according to the effects of different sub-fractions on cell viability they investigate two sub-fractions that highly affected neoplastic cell viability. Anti-proliferative and apoptotic effect is investigated in different cells. The study is of relevance in terms of novelty for the effect of sub-fractions in general but also for specific sub-fractions (IV and VI). There is also an effort to identify the compounds of the sub-fractions IV and VI.

As for the writing, major attention should be given to references as it is suspected that some error happened in the ordering phase.

I would recommend this study for publication. Nevertheless, a major point needs better clarification in terms of ethical considerations.

Major Revisions

- Although informed consent was given by the authors for using human samples, it is not clear if it was approved by an ethical committee. Clearly confirm in section 4.6 if your study needed ethical approval or not, and which bodies gave that approval or dispensed the need where appropriate (clearly specify the name of the ethics committee). For more information consult the Helsinki Declaration.

- Please explain why do you cite reference 48 as a recent work and why do you say this reference mentions the EC50 of 20 ug/ml as a reference value for potential anticancer agents.

- Please explain how do you contribute for reference 57, 58 as it is mentioned in text in lines 324-326.

Minor Revisions

General

  • Abstract should be improved in terms of writing and should to become more focused (e.g., what was the aim of this study?).
  • Along text, authors refer to I-X but forget to mention they are fractions or sub-fractions, please correct for a better understanding. Text should be improved to avoid natural speaking terms (e.g., anyway). Along the text authors refer the effects or the active sub-fractions but it is not clear what is the scientific meaning of “effects” or of “active”

Section of results

  • Rephrase sentence in line 87, what is “It”?
  • Visual legend in Figure 1,2,3 should be corrected – Did you test chloroform or extract in chloroform?
  • Line 167 – “evident” is a strong word when looking for figure 4a/b, please rephrase it.
  • Line 171 – The authors comment on the levels of cPARP at 24h and 48h, it is suggested to do same reason for Bcl2.
  • Line 172 – Following the comment on different levels for PARP at 24h and 48h it is suggested to add a comment about the difference in Bcl2 levels at 24h and 48h.
  • Figure 7 – The authors should specify the number of experiments done.
  • Figure 8 – The Y axis should mention what is it referring to.

Section of discussion

  • Discussion is too long and not focused on the article results. Reference to figures in results is missing. Initial text could be better aligned as introduction or methodology. The authors choose fraction IV and VI for a deeper study, please add the reason behind not using fraction V as it is the intermediary between IV and VI.
  • Line 268 – authors would like to substitute only to first having in mind future readers and not only the immediate ones.
  • Line 272 -the authors write effects but is a too vague term, please improve the paragraph writing.
  • Line 298 – please explain which are athe “other signaling pathways” or rephrase it.
  • Line 300 – the authors should explain what they meant by “lower concentrations and at shorter exposure time” and in which figure the reader can observe this effect. Also the respective figure should be improved adding the control concentrations as it is not clear for the reader.
  • Line 307 – Please explain where the value of EC50 of 20 ug/ml came from.
  • Line 329 the authors refer both fractions increase G0/G1/decreased G2/M phase but this is not in accordance to the line 174 in results referring fraction IV increase G0/G1 phase/decreased G2/M only in KG-1 cells. Please improve and refer figure to observe.
  • Line 331 to 337 should come after line 326 as it is referring also to apoptosis experiments.
  • Line 341 – improve “In agreement to us”

Section of methods

  • Please specify the method for TPC, TFC and TTeC as some details like the representation of units are advisable for text comprehension. The same for the ABTS method.
  • In section 4.8 explain why you used different concentrations of fractions for KG1 and for MV4-11.
  • Line 480, section 4.9 instead of “paragraph” you may want to say “section”.
  • Improve figures according to suggestions at discussion suggestion.

Reviewer 3 Report

In this manuscript, D. Lamorte et al. analyzed the chloroform fraction prepared from Azorella glabra Wedd. (AG) and sub-fractions prepared from this initial chloroform fraction. The authors focused on the anti-cancer activities of the chloroform fraction and sub-fractions with respect to acute myeloid leukemia, using two cell lines, KG1 and MV4-11, and three patient specimens. They found that subfractions IV and VI were the most potent inhibitors of cell viability, increasing apoptosis and cell cycle-arrest. They then used liquid chromatography-tandem mass spectrometry to estimate the chemical components in sub-fractions IV and VI. This study is interesting and well-conceived and may benefit AML patients in the future. Comments that the authors should address are as follows:

Please clarify certain points in the abstract: the three methods used to evaluate antioxidant activity, the use of two AML cell lines to access viability, apoptosis and cell cycle progression (only viability was measured in healthy cells).

Please add the graphs/data used to determine the EC50 values as supplementary figures.

The authors appear to be confused with respect to antioxidant effects and cancer cells. Antioxidant compounds benefit not only healthy cells but also cancer cells. One certainly does not want to treat cancer cells with antioxidants to enhance their fitness. Indeed, different types of cancer cells have mechanisms in place to combat oxidative stress. This concept was not apparent in the Results pertaining to figure 2/ Table 2 or in the Discussion.

It appears that IV and VI have a more pronounced effect on MV4-11 compared to KG1 with respect to cell viability and cell cycle-arrest. Could the authors speculate on this observation based on the genetic background of these cells?

In Figure 5, there are 24h and 48h specimens, but only one actin loading control. There should be an actin loading control for each time point. The visual band intensities do not appear to match the graph of the relative expression levels. For example, KG1 24h CTRL versus IV does not appear to be a four-fold difference. Please include Materials and Methods details regarding how western blots were quantified.

Some of the data indicate less of an effect at 48h. Could the authors comment on the stability/sustained bioactivity of IV and VI?

The AML specimens show high intrinsic apoptotic activity, which the authors should mention. Is this effect to due to stress-induced apoptosis when the cells are transiently cultured ex vivo? Based on this observation, the authors should provide additional details on how the patient cells were cultured. For example, how long were they maintained in culture prior to being analyzed?

On line 319, please change the wording from “induce” to “increase” or “enhance” apoptosis, as IV and VI appear to have a mild effect on apoptosis.

Round 2

Reviewer 1 Report

   The revised manuscript has been improved according to the reviewers' comments.